# Antifungal Hybrid Graphene–Transition-Metal Dichalcogenides Aerogels with an Ionic Liquid Additive as Innovative Absorbers for Preventive Conservation of Cultural Heritage

**DOI:** 10.3390/ma17133174

**Published:** 2024-06-28

**Authors:** George Gorgolis, Maria Kotsidi, Elena Messina, Valentina Mazzurco Miritana, Gabriella Di Carlo, Elsa Lesaria Nhuch, Clarissa Martins Leal Schrekker, Jeniffer Alves Cuty, Henri Stephan Schrekker, George Paterakis, Charalampos Androulidakis, Nikos Koutroumanis, Costas Galiotis

**Affiliations:** 1Institute of Chemical Engineering Sciences, Foundation of Research and Technology-Hellas (FORTH/ICE-HT), Stadiou Street, Platani, 26504 Patras, Greece; 2Department of Chemical Engineering, University of Patras, 26504 Patras, Greece; 3Institute for the Study of Nanostructured Materials (ISMN), National Research Council (CNR), SP35d, 9, 00010 Montelibretti, Italy; elena.messina@cnr.it; 4Department of Energy Technologies and Renewable Sources, Italian National Agency for New Technologies, Energy and Sustainable Economic Development (ENEA), Via Anguillarese 301, 00123 Rome, Italy; 5Laboratory of Technological Processes and Catalysis, Institute of Chemistry, Federal University of Rio Grande do Sul, Av. Bento Gonçalves 9500, Porto Alegre 91.501-970, RS, Brazil; 6Skeletal Biology and Engineering Research Center, Department of Development and Regeneration, KU Leuven, O&N1, Herestraat 49, PB 813, 3000 Leuven, Belgium

**Keywords:** cultural heritage, volatile organic compounds, graphene, transition-metal dichalcogenides, aerogels, ionic liquids

## Abstract

The use and integration of novel materials are increasingly becoming vital tools in the field of preventive conservation of cultural heritage. Chemical factors, such as volatile organic compounds (VOCs), but also environmental factors such as high relative humidity, can lead to degradation, oxidation, yellowing, and fading of the works of art. To prevent these phenomena, highly porous materials have been developed for the absorption of VOCs and for controlling the relative humidity. In this work, graphene and transition-metal dichalcogenides (TMDs) were combined to create three-dimensional aerogels that absorb certain harmful substances. More specifically, the addition of the TMDs molybdenum disulfide and tungsten disulfide in such macrostructures led to the selective absorption of ammonia. Moreover, the addition of the ionic liquid 1-hexadecyl-3-methylimidazolium chloride promoted higher rates of VOCs absorption and anti-fungal activity against the fungus *Aspergillus niger*. These two-dimensional materials outperform benchmark porous absorbers in the absorption of all the examined VOCs, such as ammonia, formic acid, acetic acid, formaldehyde, and acetaldehyde. Consequently, they can be used by museums, galleries, or even storage places for the perpetual protection of works of art.

## 1. Introduction

The air quality to which cultural objects are exposed is highly important for maintaining the appropriate conditions regarding the conservation of an artwork collection. In museums, galleries, and archives, the construction and decoration materials of the building and the technical equipment itself can be important sources of harmful chemical emissions that interact with the artworks exhibited or stored in closed environments [1,2]. The interactions between these harmful chemical substances, which are transferred by air, and the materials of art objects occur in different ways, mainly induced by temperature and relative humidity. Along with the aesthetic effect of pollutants and dust, such reactions cause material degradation and eventually result in disfiguring phenomena on the surfaces of the objects. Extra attention should be paid to containers, such as display cases, storage crates, cabinets, and drawers, where the emission of volatile chemical compounds from constituent materials can affect very badly the containing objects. In all the aforementioned enclosures, the air exchange amount is constrained in the sense that the local emissions will be easily accumulated in the closed environment until reaching concentration levels able to cause alterations to the works of art. Current museum guidelines consider that well-designed display cases are not only designed to provide an object’s visibility and physical protection but are also tools for the primary environmental control of the objects [3].

Chemically, the volatile compounds originating from display case materials and other containers used for the needs of museums are mainly organic and categorized as volatile organic compounds (VOCs) [4]. VOCs can be divided according to the boiling points at standard atmospheric pressure of 1 atm of their emissions. Thus, very volatile organic compounds (VVOCs) show boiling points <100 °C, volatile organic compounds (VOCs) are those with boiling points up to 250 °C, and semi-volatile organic compounds (SVOCs) stand for boiling points >250 °C, up to 380–400 °C [5]. These substances can be either present inside the materials as residues from their synthesis or otherwise can be produced after chemical reactions (oxidations and other degradation processes). Some of the most commonly reported VOCs that are detected inside museums are acetaldehyde, formaldehyde, acetic acid, formic acid, and ammonia [6,7,8,9,10]. The control of VOCs inside museums is one of the most effective preventive conservation methods [11], although it is hard to achieve a delay of the more invasive techniques of conservation of artworks [12]. Museums trying to keep their environment stable use various porous materials in their facilities and inside storage crates [8]. Currently, these materials are not so effective and have a limited operational time.

Except for VOCs, mold and fungus growth is one of the least controllable parameters that cause aesthetic and structural alterations to artworks and historical monuments [13]. The fungal contamination of art objects is mostly airborne, with significant seasonal variations. Water availability and high temperature are the most significant factors favoring mold growth [14]. Aside from deteriorative fungi, which affect only the objects, pathogenic fungi can cause health issues to people working with or visiting the contaminated objects [15]. The reason why the art objects are so appealing for fungal thriving is that artworks contain a diversity of organic materials, namely proteinaceous materials (like eggs, bovine milk, animal skin, and bones), polysaccharides, and oils, which in total, comprise the perfect source of nutrients for microorganisms [16]. Within this context, easel paintings of Giorgio Martini from the late nineteenth century showed the attack of different species of microorganisms at different parts of the paintings [16]. For example, *Cladosporium* and *Ulocladium* species, which produce cellulolytic enzymes, were responsible for the damage of canvas, and other species like *Aspergillus* and *Penicillium* attacked the paint binder and caused chromatic alterations and detachment of the support. Fabrics are another category that are mostly affected by microorganisms. Fabrics consist mainly of fibers, and based on the origin of the fibers, they can be divided into natural and chemical [17]. Plant fibers like cotton, linen, hemp, and jute or animal fibers like wool, silk, or leather are considered natural, while chemical fibers consist of modified natural or synthetic molecules like viscose, polyester, acryl, and PVC based [18]. Some microorganisms that have been reported to colonize fabrics of significant historical value are Arthrobacter, Microbispora, Sporocytophaga, Cellulomonas, Bacillus, Cellvibrio, Clostridium, Cytophanga, Pseudomonas, Nocardia, Streptomyces, Aspergillus, Chaetomium, Mnemoniella, Stachybotrys, Verticillium, Penicillium, Mucor, Trichoderma, Myrothecium, Rhizopus, Alternaria, Fusarium, Aureobasidium, and Cladosporium [19]. Mold inside the museums can be prevented by complying with and following the revised environmental guidelines that accept relative humidity between 40–60% for more sustainable storage [20]. Also, conservators and mycologists should collaborate to establish targeted detection and prevention practices for heritage repositories.

Graphene-based aerogels have caught scientific attention for being highly effective in VOC absorption [7,8,9]. Thanks to their unique structure and high surface area, such materials have been used as superior absorbers of toxic pollutants [21]. Graphene is a hydrophobic material, and its theoretical specific surface area is equal to 2600 m^2^/g [22]. Hence, it possesses remarkable adsorption ability for hydrophobic organic compounds and can be exploited as an excellent absorber for air purification. The precursor material for such structures is graphene oxide (GO), which is a sheet of graphene with carboxylic groups at its edges and epoxide/phenolic hydroxyl groups on its basal plane. Reduced graphene oxide (rGO) can be obtained after chemical treatment or thermal annealing, eliminating such functional groups on GO [21]. Compared with GO, rGO can reach a higher absorption capacity for aromatic pollutants thanks to its lower oxygen content, higher hydrophobicity, and higher surface area. Chen et al. [23] examined the absorption of *m*-dinitrobenzene, nitrobenzene, and *p*-nitrotoluene onto GO and rGO, showing that rGO nanosheets have higher absorption capacities for nitroaromatic compounds. Apart from hydrophobicity, π–π interactions also contribute to the strong adsorption of organic molecules onto graphene-related materials [24]. Wang et al. [25] examined the adsorption of phenolic compounds on rGO and concluded that the adsorption depends on π–π interactions between the aromatic molecules and rGO. The degree of rGO reduction and the chemical structure of the phenols can affect the π–π interactions.

Another category of two-dimensional materials in tandem with graphene is the transition-metal dichalcogenides (TMDs), which are known to be able to adsorb environmental pollutants [26,27,28]. The fascinating characteristics of the TMDs, like the layered structure, tunable bandgap, and unique optical, thermal, and electrical properties, are the main reasons why the potential of these two-dimensional materials is explored for pollution prevention [26]. Graphene can be combined with TMDs and create three-dimensional macrostructures like aerogels with enhanced properties. The as-resultant structures are usually named as ‘*hybrid*’ aerogels [8,29]. Worsley et al. [29] have reported molybdenum disulfide (MoS_2_)/graphene aerogels with a high surface area of ~700 m^2^/g and an electrical conductivity of 112 S/m. Also, Zhu et al. have explored graphene/MoS_2_ aerogels for the absorption of water-soluble organic contaminants [30]. The 3.2 wt.% polydopamine addition resulted in a composite structure with small MoS_2_ nano-crystallites homogeneously dispersed over the graphene surface without aggregation.

Ionic liquids (ILs) are a very interesting category of materials and consist of a discrete cation and anion pair. The research on these salts with low melting temperatures was intensified thanks to their tunable properties, high ionic conductivity [31], good thermal stability [32], low flammability [33], negligible vapor pressure [34], and tunable polarity and surface activity [35]. Regarding their biological activity, various ILs have been studied as potential molecules for antitumoral [36] and antimicrobial agents, due to their relatively low toxicity [37,38]. Such biological activity of the ILs strongly depends on the charge, size, alkyl chain, electronegative group, charge distribution on the ions, and small changes in the shape of covalent bonds with protein or micro-molecules interaction, and their thermal properties as well [39,40,41]. In the case of the imidazolium ILs, 1-hexadecyl-3-methylimidazolium chloride (C_16_MImCl) has been identified as a strong antimicrobial [37,38] and antitumoral agent [36]. The incorporation of this IL in materials represents an effective strategy for the preparation of biologically active materials, including poly(L-lactide)- [42] and high-density polyethylene-based [43] biomaterials, and calcium phosphate-based bionanocomposites [44]. The advantages of using ILs in such processes can be listed as the following. (1) It is considered an easily applicable and cost-efficient way to create tunable, diverse libraries of biologically active compounds with a countless number of combinations of anions and cations. (2) ILs result in control of the ion formation in a solution and the adjustment of the solvation properties in water and biological fluids to provide a reliable solution for solubility and bioavailability. (3) Their hydrophobic/hydrophilic properties, nature of the ionic core, covalent/ionic binding, linker size, and characteristics can be finely tuned via simple organic synthesis procedures. On the other hand, some challenges that are currently faced are the following. (1) Mechanisms of action of many biologically active compounds with ionic nature are not well understood and require further examination. (2) Systematic comparative studies on different types of IL-based drug development systems have not yet been reported, since most of the reported works focus on specific types of IL systems. (3) The relationship between the molecular structure and nano-/microscale arrangement of molecular properties into self-organized structures is at the first stages of understanding [45].

Our research consortium has recently reported on graphene-based aerogels with an IL (C_16_MImCl) additive, exhibiting anti-fungal and VOC absorption properties [7]. Several IL contents were evaluated, and the graphene aerogel containing 10 wt.% IL was found to have the best anti-fungal activity, preventing the aerogel contamination with *Aspergillus niger*. When exposed to a VOC-saturated micro-environment, this aerogel is highly suitable for the absorption of acetaldehyde, formic acid, acetic acid, and formaldehyde. Since the combined anti-fungal and VOC absorption properties are prerequisites for bringing preventive conservation to a higher level, herein, graphene was combined with MoS_2_, tungsten disulfide (WS_2_), and C_16_MImCl to prepare ‘*hybrid*’ absorbent materials for enhanced selective absorptions of pollutants that present, at the same time, anti-fungal activity. It is strongly believed that, apart from the context of the preventive conservation of cultural heritage, these results are of much general importance, including the chemical industry and research laboratories.

## 2. Experimental Procedure

For the materials, an aqueous solution of GO was prepared by the modified Hummer’s method [45,46] and was subsequently diluted in water to obtain a concentration of 1 mg/mL. The commercial WS_2_ (Sigma-Aldrich, Steinheim, Germany, 2 μm, 99%) and MoS_2_ (Sigma-Aldrich) powders were used as received. C_16_MImCl was synthesized as described in the literature [47,48,49].

For the preparation of the ‘*hybrid*’ graphene—TMDs aerogels, the approach presented by Hong et al. [50] for the simultaneous assembly and reduction of GO (the steps for the synthesis of neat rGO aerogel are described in the Appendix A) was adapted. The freeze-drying technique was exploited for solvent sublimation and obtaining the aerogels. Ethanol–water (50/50% *w*/*w*) solutions of 1 mg/mL of 2D material bulk platelets (MoS_2_ and WS_2_) were prepared and bath-sonicated for 10 min. The GO solution was mixed with one of the two 2D materials in contents of 90/10%, 70/30%, and 50/50% *w*/*w* and stirred for 10 min, and then, the same steps were followed as for the preparation of the neat GO (see Appendix A). For the ‘*hybrid*’ aerogels with the IL additive, at the stage of the reducing agents’ addition (H_3_PO_2_ and I_2_, see Appendix A), the amount of C_16_MImCl was added based on the IL:GO weight ratio of 1:10 *w*/*w*. The as-prepared samples were correspondingly coded as rGO, rGO10IL, rGO10IL/MoS_2_, and rGO10IL/WS_2_.

For scanning electron microscopy (SEM), SEM images were obtained using a LEO SUPRA 35VP with a maximum resolution of 1.5 nm and 2 nm at high and low vacuum, respectively.

For Raman spectroscopy, Raman spectra were collected using a Renishaw InVia Raman Spectrometer with a 1200 grooves/mm grating for the 785 nm laser excitation and several lenses, such as 20×, 50×, or 100×. The power of the laser beam was kept lower than 1 mW to avoid heating of the specimens.

For the attenuated total reflectance Fourier transform infrared spectroscopy (ATR-FTIR), ATR-FTIR spectra were collected using a Nicolet iS50 spectrometer (Thermo Fisher, Waltham, Massachusetts-United States) equipped with an ATR accessory. The measurements were recorded using a germanium crystal cell using, typically, 32 scans at a resolution of 4 cm^−1^. No ATR correction has been applied to the data. The range of wave numbers within which the measurements were performed was equal to 450–1650 cm^−1^. 

For X-ray diffraction (XRD), a Bruker D8 Advance X-ray diffractometer was used for performing the XRD measurements. The specimens were examined under ambient conditions.

For X-ray photoelectron spectroscopy (XPS), the surface-analysis measurements were performed in a UHV chamber (P ~5 × 10^−10^ mbar) equipped with a SPECS Phoibos 100-1D-DLD hemispherical electron analyzer and a non-monochromatized dual-anode Mg/Al X-ray source for XPS. The XP Spectra were recorded with MgKa at 1253.6 eV photon energy and an analyzer pass energy of 10 eV giving a full width at half maximum (FWHM) of 0.85 eV for the Ag3d_5/2_ line. The analyzed area was a spot with a 3 mm diameter. The atomic ratios were calculated from the intensity (peak area) of the XPS peaks weighted with the corresponding relative sensitivity factors (RSF) derived from the Scofield cross-section, taking into account the electron transport properties of the matrix (the inelastic mean free path (IMFP). *λi* and the elastic-scattering correction factor *Q* depend mainly on the corresponding electron kinetic energy (KE)) and the energy analyzer transmission function. For spectra collection and treatment, including fitting, the software SpecsLab Prodigy with version No 4.12.0r49869 (Specs GmbH, Berlin, Germany) was used. The XPS peaks were deconvoluted with a sum of Gaussian–Lorenzian peaks after a Shirtey-type background subtraction.

For the volatile organic compounds (VOCs) absorption tests, the gas absorption tests were conducted under static conditions in a closed glass desiccator with an excess of pollutant using a saturated vapor stream at room temperature (Appendix A), as described also in detail in the Appendix A. The absorption capacity [51] of the prepared graphene aerogels was determined by calculating the percentage of weight change:$$A\%=\frac{last\;weight\;measurement-initial\;weight\;measurement}{initial\;weight\;measurement}\times 100\%$$

Since the graphene-based aerogels reached the saturation point in the volatile gas absorption, which corresponds to the maximum weight change, the weight of the aerogels was stabilized. For the establishment of a baseline for the saturation of the aerogels, after two consequent gravimetric measurements similar to the maximum observed value, the samples were considered saturated and were submitted to the regeneration process. The regeneration was carried out by drying the absorber and removing the absorbed pollutant above its boiling temperature. The regeneration of the hybrid aerogels after their saturation in the VOCs absorbance was performed by heating them (60 °C) with a common electric hair dryer for 24 h (Appendix A), as published elsewhere [52]. Alternatively, overnight drying at 120 °C in an oven with a vacuum can be also efficiently used. All samples were tested for three absorption–desorption cycles.

For the anti-fungal tests, these tests were conducted in triplicate, placing the aerogels in Petri dishes with a malt agar extract as the culture medium, followed by incubation in the dark at 25 °C. Finally, the fungal growth and, thus, the percentage of Petri dishes colonized by *Aspergillus niger* were verified after 24 h and quantified by image analysis using ImageJ software with version No 1.8.0.

## 3. Results and Discussion

The two-dimensional materials were kept in bulk form to maintain their ability to trap gaseous pollutants [53], avoiding also the elevated cost of the exfoliation process. Ethanol was added to ensure a good dispersion of the bulk 2D materials, which also significantly stabilized the solutions. It is reported elsewhere that a mixture of 50/50% *w*/*w* of ethanol–water results in good dispersion of two-dimensional nanosheets and that ethanol can be removed from hydrogels during the freeze-drying process [53,54]. When only water was used, the platelets tended to separate from the solution, and even though gelation still occurred, there was a non-uniform distribution of the platelets in the formed aerogels. The density of the prepared aerogels was found to lie in the range of 13.9–23.4 mg/cm^3^, which is within the desirable range of values, without significant variation between samples. In fact, these values (less than 30 mg/cm^3^) are a prerequisite for classifying the 3D porous structure as an aerogel [30].

Since the 2D materials do not contain reactive groups on their surfaces, they were randomly dispersed in the starting GO solution. During the formation of the conjugated network of rGO sheets, the bulk nanosheets of 2D materials got wrapped in the 3D rGO structure, which served as a rigid template for the final 3D porous hydrogel [8]. The result was a 3D rGO foam with two-dimensional nanosheets randomly attached to its pores, as shown in the SEM images of the produced samples. To examine the size and structure of the pores of the prepared aerogels, SEM images for all the IL-free samples with various mixed ratios were also acquired (Figure 1). The platelets of the 2D materials are clearly distinguished and seem to be either attached to the surface of the rGO layers or wrapped around them, creating a robust macro-scale interconnected porous network. Also, the magnitude of the pores tended to increase for the samples with a higher content of the nanosheets, and the prepared aerogels are considered mainly macro-porous structures (because the size of pores exceeded 50 nm). Also, some meso-pores [55] are detectable. It is known that the porosity of carbon aerogels can be controlled by the size and shape of the sheets and the concentration of the precursor GO solution [56]. Since the same GO stock solution was used for the preparation of all the samples, this observation can be explained by considering the concentration of the initial GO solutions for each set of aerogels (90/10, 70/30, and 50/50 rGO/nanosheets). In the preparation of the pure rGO aerogel, the concentration of the used GO was 1 mg/L, while for the rGO/TMD hybrids, the concentration of GO was lower. Even though the starting GO solution for hybrid aerogels had lower GO content, the final volume of these aerogels was the same as that of the neat rGO [8], which means that the porous network inside them should be less dense.

The survey XPS scan (Figure 2) of the rGO/MoS_2_ *50/50* aerogel showed, except for the signals of the C, O, and P atoms related to rGO, a spin-orbit doublet at 229.2 eV (Mo3d_5/2_) and 232.4 eV (Mo3d_3/2_), which is attributed to the Mo^4+^ of MoS_2_. Furthermore, two S_2p_ peaks at 162.2 eV (S2p_3/2_) and 163.3 eV (S2p_1/2_) were observed, assigned to S^2−^ of MoS_2_ [29]. The XPS survey spectrum of the rGO/WS_2_ *50/50* aerogel (Figure 3) demonstrated the coexistence of C, O, W, and S atoms. The binding energy of the S_2p_ peak due to S^2−^ appears at 162.2 eV and 163.4 eV, corresponding to S2p_3/2_ and S2p_1/2_, while W4f_7/2_ and W4f_5/2_ are observed at 32.5 eV and 34.7 eV, respectively, indicating the existence of W^4+^ in WS_2_ [57]. The sulfur-to-tungsten ratio was calculated to be 2.2, which is slightly S-rich. Additionally, to find the reduction level of the hybrid aerogels with TMDs and compare it to the reduction level in the pure rGO sample, the C:O atomic ratio was calculated for each specimen, subtracting the oxygen concentration due to the presence of P_2_O_5_ (resulting from the addition of hypophosphorous acid (H_3_PO_2_), see Appendix A). These levels were equal to 10.4 and 8.3 for the rGO/MoS_2_ and rGO/WS_2_ samples, respectively. The calculated C:O ratio indicates that the higher the percentage of two-dimensional material contained in a sample, the lower its reduction level. This observation was further assessed by Raman measurements, which are presented below.

Figure 4 shows the results of XRD and Raman measurements for the hybrid samples. Both analyses proved the presence of rGO and TMDs in the hybrid aerogels. The XRD plot of a neat rGO aerogel shows a broad peak at 2θ = 26°, which corresponds to the (002) plane of graphite [58]. For the rGO/TMD *50/50* hybrid aerogels, the obtained XRD spectra confirm the crystalline nature of MoS_2_ and WS_2_ [56,58]_,_ and a lower reduction level of GO as the peak of rGO shifted to lower 2θ values compared with the 26° of neat rGO [59]. The Raman spectra of a neat rGO aerogel comprise several frequency bands, each one assigned to a specific structural configuration [60]. For the main Raman peak, the so-called G peak is located at ~1590 cm^−1^ and is related to the E_2g_ mode that corresponds to the in-plane stretching of C=C bonds [61]. Another prominent Raman peak that has been associated with the presence of disorder has been termed the D peak, and it is present at about 1356 cm^−1^. It corresponds to a breathing vibrational mode, and its intensity is associated with the number of lattice defects or discontinuities, such as flake edges [51,61]. The two main Raman peaks of GO, G and D, are very sensitive to small sp^3^/sp^2^ ratio changes. Also, the intensity ratio of these two peaks I_D/G_ is correlated to the changes in the sp^3^/sp^2^ ratio and can be used to confirm the reduction of GO. From the calculated ratio of D/G peaks, it is evident that the highest reduction occurred in the neat rGO aerogel, while the lowest ones occurred in hybrid aerogels with the highest 2D materials content. The main Raman peaks found for the MoS_2_ aerogel, except the peaks of rGO, are those at 408 cm^−1^ (first-order mode A_1g_) and 373 cm^−1^ (second-order mode E_2g_), and for the WS_2_ aerogel, there are those at 420 cm^−1^ and 355 cm^−1^ [62,63]. These peaks’ positions are in good agreement with similar values from the literature for nanoparticles that are composed of multiple layers [29].

For the hybrid aerogels (rGO/MoS_2_ *50/50* and rGO/WS_2_ *50/50*) with the IL additive, the SEM, XRD, and Raman measurements showed no significant differences in comparison with the samples without IL. The XPS measurement on an rGO/IL aerogel, prepared under the same conditions but without TMD, detected the existence of nitrogen (N) with a calculated atomic concentration (%) equal to 2.62 (Appendix A).

### VOCs Absorption

The prepared hybrid aerogels were tested for sorption of formaldehyde (CH_2_O), acetic acid (CH_3_COOH), formic acid (CH_2_O_2_), acetaldehyde (CH_3_CHO), and ammonia (NH_3_), as well as water vapor (humidity), which are the most common pollutants in places where artworks are exhibited.

The regeneration ability was examined, while the significance of this process lies in the necessity of recyclability of the 3D material from an environmental, economic, and practical perspective. Ideally, absorbent material is desired to maintain its absorption ability after many regeneration cycles. From the absorption results summarized in Figure 5 and Figure 6, the incorporation of TMDs in rGO aerogels favored the selective absorption of ammonia, while the absorption capacity increased with increasing TMD content. Thus, the most efficient ammonia absorbers were the hybrid aerogels with the highest amount of MoS_2_ and WS_2_, namely the *50/50* samples. For the assessment that the higher increase in weight of the samples with TMDs was not due to enhanced humidity absorption, similar sets of samples were tested under 55% RH (inside a desiccator with a saturated aqueous salt solution [64]), and did not show any weight increase. This means that the weight increase of the aerogels was caused only by ammonia sorption. Meanwhile, for the cases of formaldehyde, acetic acid, formic acid, and acetaldehyde, the opposite trend was observed for the aerogels containing MoS_2_, with the rGO/MoS_2_ *90/10* sample being the most efficient absorber. At the same time, the hybrid aerogels with WS_2_ showed a dramatic decrease in absorption capacity for the examined VOCs (apart from ammonia), particularly for the *50/50* sample, and a 40% decrease for both formaldehyde and acetic acid was recorded. An important note is that the humidity absorption at 75% RH was insignificant, in the range of 4–6%, while at 99% RH, the absorption capacity was decreasing with the increasing TMD content (rGO/MoS_2_ *90/10*: 32.4%, rGO/MoS_2_ *70/30*: 22.2%, rGO/MoS_2_ *50/50*: 6.7%, GO/WS_2_ *90/10*: 31.8%, rGO/WS_2_ *70/30*: 16.6%, and rGO/WS_2_ *50/50*: 5.6%, compared with the neat rGO aerogel). This behavior can be ascribed to the hydrophobic nature of TMDs. As a result, water molecules are physiosorbed with a low degree of charge transfer on TMDs [65]. When the performance of the proposed ‘*hybrid*’ aerogels of this study is compared with the corresponding ones of neat rGO aerogels (Appendix A) [7], the absorbance values are found to be two-to-three (2–3) times lower on average for all the examined VOCs, apart from ammonia. For the case of ammonia, the neat rGO aerogel shows an absorption of only 84% (Appendix A), while the corresponding values for rGO/MoS_2_ and rGO/WS_2_ are equal to 273% and 232%. This difference of one order of magnitude in the absorption performance reveals the selectivity for ammonia of the ‘*hybrid*’ aerogels.

There is extended research on ammonia detection using TMDs, so their selective absorption of ammonia after incorporation in rGO aerogels was maintained [64,65]. In the case of neat rGO aerogels, the absorption of ammonia takes place only on oxygen-containing functional groups, such as epoxy and hydroxyl, through hydrogen bonding [66,67], while when adding MoS_2_ and WS_2_ into the rGO structure, practically active adsorbing sites with selectivity to ammonia are added [68,69,70]. Owing to the existence of a lone pair of electrons, ammonia behaves as a charge donor to provide electrons to get physiosorbed on TMDs. Due to the strong electronegativity of the elemental sulfur layer of the TMD layers, NH_3_ molecules are easily adsorbed to the edge sites of the few-layered and bulk TMDs and transfer electrons to the TMD layers to form NH_4_^+^. Thus, NH_4_^+^ ions adhere to the edges of two adjacent elemental sulfur layers and expand the interlayer space. Subsequently, more NH_3_ molecules enter deeper inside, expand the interlayer space, and eventually fill the entire interlayer space [64,65,71,72,73]. Another observation from the absorption-capacity results (Figure 5 and Figure 6) is that MoS_2_ samples exhibited higher ammonia absorption relative to the WS_2_ samples. This can be related to the larger radius of W^4+^ in WS_2_, resulting in a weaker bonding ability to ammonia than for Mo^4+^ in MoS_2_, which is also supported by the calculated absorption energy for ammonia of −216 eV for WS_2_ and −250 eV for MoS_2_. That practically means that the desorption of ammonia from WS_2_ becomes easier compared to that from MoS_2_ at room temperature. This phenomenon also has been found to contribute to the better recovery feature of the WS_2_-based ammonia sensors, when compared to the MoS_2_ ones [53].

Regarding the other VOCs, the addition of TMDs into rGO aerogels resulted in lower absorption capacities, which has been also reported elsewhere [53]. TMDs cannot interact with VOCs through *π–π* stacking or hydrogen bonding, but only via electrostatic attraction forces. The flakes of TMDs (*p*-type), which were dispersed in the three-dimensional pores of rGO aerogels, interact with some active adsorption sites, oxygen-containing groups (*n*-type), and form *p–n* junctions, leading to less active sites of rGO aerogels for absorbing VOCs. Furthermore, the TMDs are hydrophobic, especially tungsten disulfide, and this is not favorable for attracting molecules of formaldehyde, formic acid, acetic acid, and acetaldehyde. Despite this drawback, there is a positive aspect to the absorption of formic acid under high RH. As the hydrophobicity of TMDs disfavors the competitive adsorption of water molecules, the hybrid aerogels favor the absorption of formic acid molecules. The hybrid aerogels with TMDs maintained similar absorption capacities of ammonia through all the absorption–desorption cycles (Figure 5 and Figure 6). This behavior can be attributed to the reversible interaction of ammonia with TMDs. Moreover, Raman spectroscopy revealed that ammonia promoted a further reduction of the hybrid aerogels with TMDs, which did not occur with the other examined VOCs. More specifically for the rGO/WS_2_ *50/50* aerogel, the ratio of the intensities of characteristic D and G peaks was increased from 1.01 ± 0.06 to 1.05 ± 0.016. A red shift of the G peak was detected from 1591 cm^−1^ to 1587 cm^−1^, proving the further reduction of rGO/TMDs aerogels. Contrarily, the absorption capacities and saturation times of formaldehyde decreased after regeneration. This can be justified by the fact that TMDs do not exhibit selective adsorption of formaldehyde, so all the formaldehyde molecules were attached to rGO active sites, leading to degradation. Meanwhile, the acquired Raman spectra detected no further reduction of rGO.

Although TMDs are prone to oxidation, WS_2_ is considered to be more stable than MoS_2_. When water and pollutant molecules are physiosorbed on TMDs, they can be transformed into semiconducting metal oxides (MoO_3_, WO_3_). Yet, they maintain their sensing and selectivity properties [68]. In the case of acetic acid, some fluctuations of the acquired data were observed, and these can be attributed to slight oxidation after exposure to the acid, as shown from Raman measurements. Regarding the saturation time of the samples, in general, the trend remained the same for all the VOCs except ammonia, which showed better performance.

As shown in Figure 7 and Figure 8, the addition of the IL (10% *w*/*w* related to GO) resulted in increased absorption capacities for all examined VOCs. ILs have been reported to exhibit remarkable affinities for VOCs [74]. C_16_MImCl should have induced a significant increase in hydrophobicity, enhancing the surface potential of the GA-based surfaces [75] and turning the interaction with the VOC molecules stronger [76]. Furthermore, such IL favors the formation of strong hydrogen bonds with VOCs [77]. According to molecular interactions between the VOCs and the ions of the ILs, the VOCs do have preferential adsorption near the IL interfacial region compared to the bulk region [78]. The VOC molecules have been reported to prefer to reside near the IL/VOC interface, while, also, the longer the length of alkyl substituents of the imidazolium cation is, the higher the solubility of VOCs in ILs [79]. Acetaldehyde was absorbed the most, and this could be attributed to its high volatility, which depends on its boiling point and saturation vapor pressure [80]. In addition, for acetaldehyde, there is a synergistic action of dipole–dipole and hydrogen bonding interactions with the GA surface [81].

Even though some of the aerogels showed lower performances after the first absorption–desorption cycle, the samples exhibited significantly higher absorption capacities than the other relevant reported materials [8]. Similar experiments have been performed for testing reduced graphene oxide-based porous materials as gas absorbers for various VOCs, including benzene, toluene, carbon dioxide, ammonia, acetic acid, and formaldehyde [8]. In addition to rGO aerogels, various other materials have been studied as pollutant removers, ranging from activated carbons to zeolites and from metal–organic frameworks (MOFs) to hyper-crosslinked polymeric resins (HPRs). Thanks to their large surface areas, these porous materials are capable of trapping volatile pollutants either by physical or chemical absorption [81,82]. Nevertheless, several drawbacks of commercially available porous materials have been discovered when it comes to VOC removal. For example, activated carbon cannot capture VOCs with low molecular weight like formaldehyde, and other materials may not be effective in capturing polar VOCs like ammonia [83]. Hence, the selectivity, as well as the reusability of these materials, needs improvement. Also, the manufacturing costs of the materials play a crucial role. In fact, to make the materials attractive for practical applications in museums and art galleries, where the real needs are, for thousands of display cases or storage boxes, the production costs should be as low as possible. Activated carbons, natural porous materials, and natural zeolites represent reasonable compromises between production costs and environmental impact. Indeed, activated carbons have tremendous potential thanks to their low manufacturing costs and thanks to the biomass nature of the raw materials, which could be coconut shells, walnut shells, woods, and many more. Also, some natural porous materials, such as diatomite, stellerite, and vitric tuff, have low environmental impact and could be used as VOC removers after easy activation steps. Despite this, both natural porous materials and activated carbons have low VOC absorption capacities, and even more importantly, their powdered nature makes their practical usage extremely limited (e. g. they need filters, boxes, etc.).

Some of the most common commercial absorbent materials were used for the benchmarking study, like activated carbon, silica gel, and polyurethane. These materials were dried and placed inside the desiccators with the same VOCs, and the calculated absorption capacities and kinetics are presented in Appendix A. This confirmed that commercial absorbent materials exhibit significantly lower absorption performance than graphene-based aerogels without any particular selectivity. Comparing the best commercial absorbent with the best aerogel sample for each examined VOC, the rGO/MoS_2_ and rGO/WS_2_ *50/50* samples showed about 530% more weight increase for ammonia absorption than polyurethane. Also, it is worth mentioning that the superior performance of the presented materials was achieved by using much less weight (a few milligrams) compared with the commercial absorbents (a few grams). Additionally, the commercial materials were tested for humidity absorption under the same RH conditions to examine whether their weight increase was due to water or VOC absorption. For 55% RH, polyurethane absorbed significantly less humidity than activated carbon and silica gel. So, it can be concluded that polyurethane has good selectivity for ammonia and acetic acid over water molecules. In the case of 75% RH, polyurethane again exhibited significant selectivity for formaldehyde absorption over humidity, contrary to activated carbon and silica gel. Finally, for 99% RH, none of the commercial absorbents had selectivity for formic acid absorption over humidity. Thus, it can be also deduced that the detected weight increase of the examined commercial materials after exposure to formic acid was entirely because of humidity absorption. As a consequence, this benchmarking study showed that the graphene/TMDs-based aerogels exhibit superior absorption properties and gas selectivity compared to the available commercial products.

The as-prepared ‘hybrid’ aerogels (without the addition of the IL) were put into the absorption system to investigate their interactions with VOCs by FTIR analysis, as described in the experimental procedure and in a previous work [7], followed by desorption treatments to remove the physiosorbed pollutants and evaluate their ability to capture VOCs with a stable interaction. Representative results of the absorption tests after VOC treatments are reported in Figure 9A for rGO/WS_2_ *50/50* and in Figure 9B for rGO/MoS_2_ *50/50*. For all the rGO/TMD aerogels (untreated = UTT and treated = TT), there are three important bands of the oxygen-containing epoxide functional group at ~1154 cm^−1^ (C-O-C stretching peak), ~1000 cm^−1^ (C-O-H deformation peak), and ~800 cm^−1^ (C-OH stretching peak). The band at 1567 cm^−1^ is attributed to the C=C skeletal vibration of rGO/TMD [84]. Additionally, an extra peak emerged at ~470 cm^−1^, which is characteristic of the Mo-S stretching mode of vibration, confirming the successful incorporation of MoS_2_ in the rGO/TMD (identified with a red star in Figure 9B) [85].

The doping of rGO aerogels with TMD nanosheets (MoS_2_ or WS_2_) introduced TMD-originated acidic centers and resulted in rGO with a higher content of functional groups due to a lower degree of reduction. As a consequence, the surfaces of these materials became more sensitive and selective to the attachment of ammonia molecules, which was shown in the absorption-capacity experiments (Figure 5 for rGO/MoS_2_ aerogels and Figure 6 for rGO/WS_2_ aerogels). The FTIR spectra acquired after the exposure of rGO/TMD aerogels to ammonia show the ability to capture this VOC with a stable interaction. The most visible band at 1440 cm^−1^ can be attributed to the characteristic NH deformation vibration of NH_4_^+^ (identified with an orange star for rGO/TMDs TT) [86]. The formation of NH_4_^+^ was due to ammonia hydrolysis in contact with atmospheric or adsorbed water molecules. For all the tested VOCs, the FTIR spectra of the rGO/TMD aerogels exposed to a VOC show changes compared with the spectra of their untreated equivalents. This confirms the occurrence of chemical changes in the surfaces of these aerogels after treatments with VOCs.

The antifungal properties of rGO and rGO-based hybrid aerogels have been investigated by evaluating their capacity to inhibit the proliferation of *Aspergillus niger*. This fungus was selected as a model microorganism, representative of those commonly detected in museums [87]. In particular, the effect of TMDs (MoS_2_ and WS_2_) on the antifungal properties of rGO aerogels (rGO/TMDs *50/50*), without or with the IL, was studied. This was evaluated by the ability of these aerogels to inhibit fungal growth in the surrounding environment. The results reported in the following Table 1 show that the addition of 10 wt.% of IL improved the antifungal property of the rGO aerogel, significantly reducing the average percentage of the Petri dish covered by the fungus, which was demonstrated in our previous work [7]. By embodying a TMD (MoS_2_ or WS_2_) in the aerogels with the IL additive, increased surface areas of the Petri dishes were covered by the fungus, although an inhibition effect was maintained. The results presented in Figure 10 show that all the aerogels were able to inhibit the fungal growth in the culture medium. Representative pictures of the Petri dishes after 24 h of incubation are shown in Figure 11, visualizing the inhibition effect of the different formulations. The obtained results clearly indicate the positive effect of IL. Although this effect was partially lost by the incorporation of a TMD (likely the result of a strong interaction between the TMD and the IL), it is worth noting that all rGO-based aerogels considered in this study were effective in preserving, to varying degrees, their surfaces and surroundings from fungal attack. The latter properties were mainly improved through the addition of IL.

## 4. Conclusions

In conclusion, macroporous rGO/TMD (MoS_2_ or WS_2_) hybrid aerogels, without or with the IL C_16_MImCl, were successfully prepared through freeze-drying as novel absorbers of VOCs with antifungal properties. These materials can be created with varying content of the examined TMDs in the rGO backbone, specifically *90/10*, *70/30*, and *50/50%* *w*/*w* (TMD/starting GO). For the rGO/TMDs *50/50* samples, SEM confirmed the macroporosity with clearly shown TMD crystals, while with the assistance of XPS, XRD, and Raman spectroscopy, the characteristic peaks of the TMDs were detected, except the already shown ones of the rGO. All hybrids showed high absorption rates with ammonia, formic acid, acetic acid, formaldehyde, and acetaldehyde, which was absorbed the most, exceeding the corresponding performances of other benchmark products, like activated carbon, silica gel, and polyurethane. As these superior performances were maintained in three cycles of absorption–desorption tests, these hybrids can be effectively reused after a simple drying process. The addition of a TMD enhanced the absorption of ammonia, providing materials for the selective absorption of ammonia with the rGO/TMDs *50/50* exhibiting the highest performance. When the rGO/MoS_2_ structure is compared with the rGO/WS_2_ one, the absorption of ammonia for the first structure is clearly higher than the latter one. FTIR spectroscopy was also used for the detection of the absorption of all the examined pollutants, while especially for ammonia, a stable interaction was deduced. Especially after the ammonia absorption, the most visible band at 1440 cm^−1^ was attributed to the characteristic NH deformation vibration of NH_4_^+^. The formation of NH_4_^+^ was due to ammonia hydrolysis in contact with atmospheric or adsorbed water molecules. This finding confirms the occurrence of chemical changes in the surfaces of these aerogels after the treatment with this VOC, rendering a spectroscopic proof of their absorbing ability. C_16_MImCl as an additive potentialized the absorption ability for all the examined VOCs, without altering the previously-reported selectivity, and the anti-fungal activity against the fungus *Aspergillus niger*, which is a model microorganism and representative of those commonly detected in museums, galleries, private collections, et al. This anti-fungal intrinsic property was evaluated by the ability of these materials to inhibit fungal growth in their surrounding environment. From the results, it is shown that the incorporation of TMDs in the aerogels with IL additive increased the surface areas of the Petri dishes that were covered by the fungus, but a significant inhibition effect was maintained. All examined formulations were able to inhibit fungal growth in the culture medium, indicating, thus, the positive effect of IL. All the aforementioned findings turn these hybrid aerogels into promising tools for the preventive conservation of cultural heritage.

## Figures and Tables

**Figure 1 materials-17-03174-f001:**
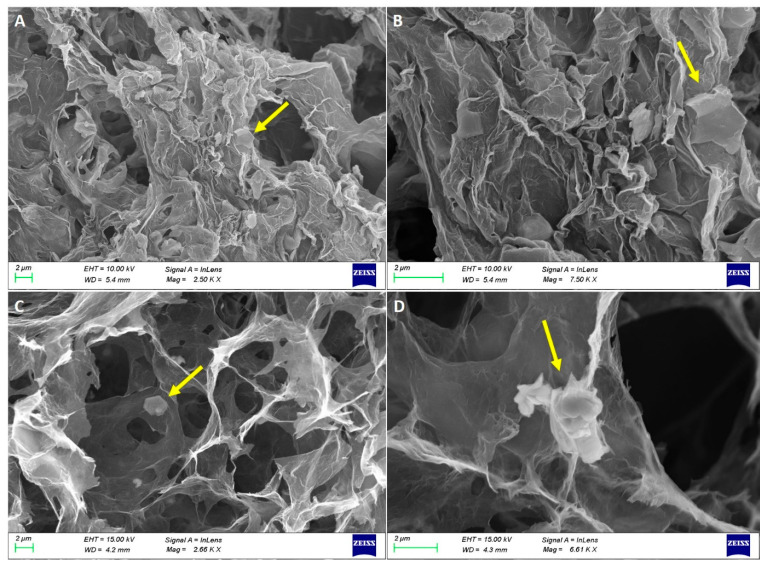
SEM images of rGO/MoS_2_ *50/50* (**A**,**B**) and rGO/WS_2_ *50/50* (**C**,**D**) aerogels with different scale bars. The yellow arrows point to flakes of MoS_2_ or WS_2_, correspondingly.

**Figure 2 materials-17-03174-f002:**
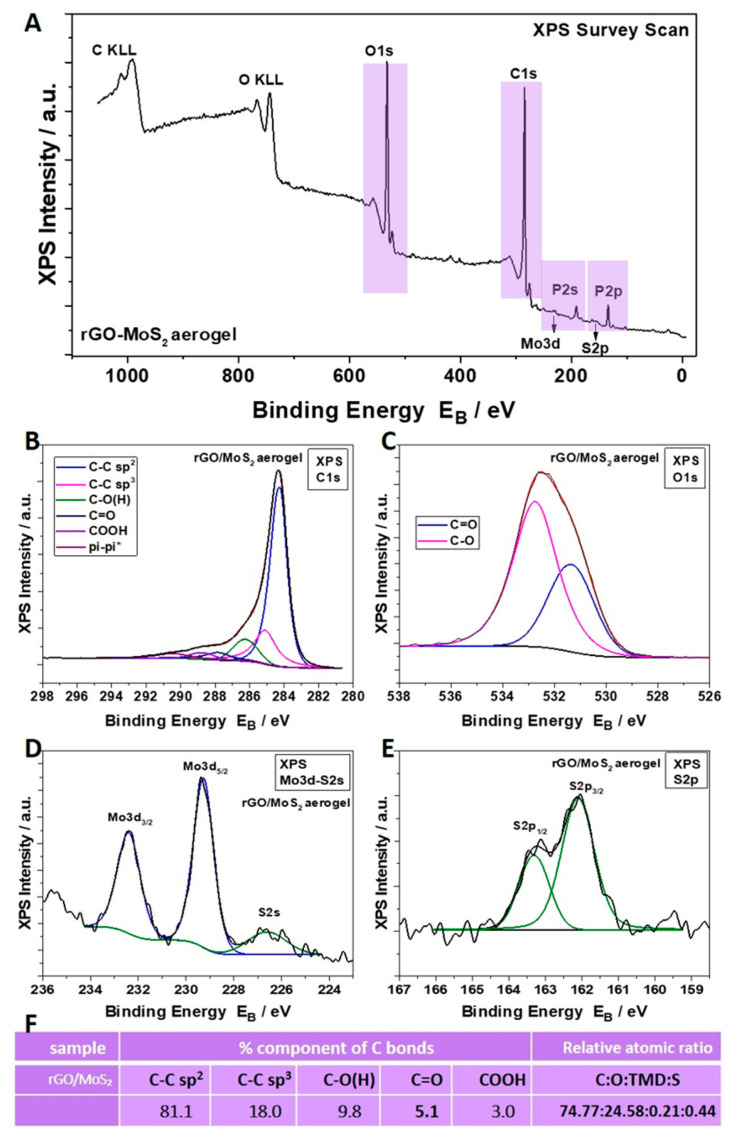
(**A**) XPS survey scan of the rGO/MoS_2_ *50/50* hybrid aerogel. Deconvolution of its C_1s_ (**B**) and O_1s_ (**C**) peaks. XPS peaks attributed to its Mo^4+^ of MoS_2_ (**D**) and S^2−^ of MoS_2_ (**E**). (**F**) Percentages of C_1s_ components derived from the C_1s_ peak deconvolution and relative atomic ratio C:O:TMD:S of this hybrid aerogel.

**Figure 3 materials-17-03174-f003:**
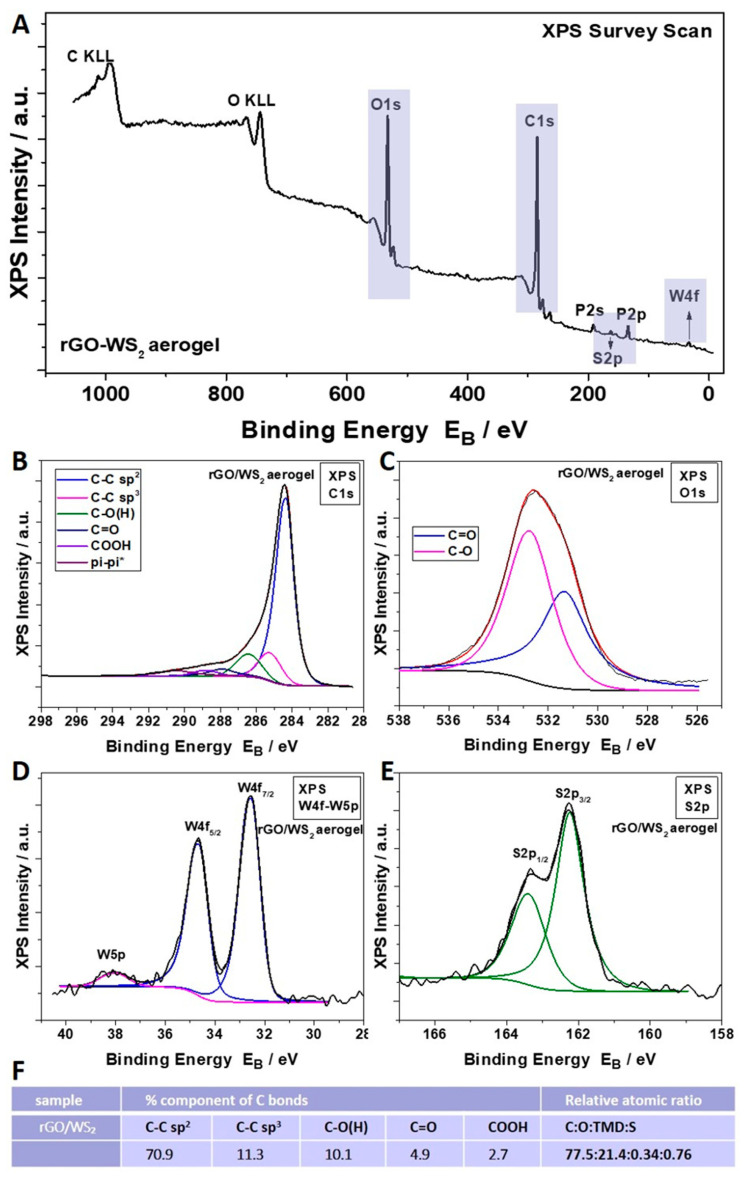
(**A**) XPS survey scan of the rGO/WS_2_ *50/50* hybrid aerogel. Deconvolution of its C1s (**B**) and O1s (**C**) peaks. XPS peaks attributed to its W^4+^ of WS_2_ (**D**) and S^2−^ of WS_2_ (**E**). (**F**) Percentages of C1s components derived from the C1s peak deconvolution and relative atomic ratio C:O:TMD:S of this hybrid aerogel.

**Figure 4 materials-17-03174-f004:**
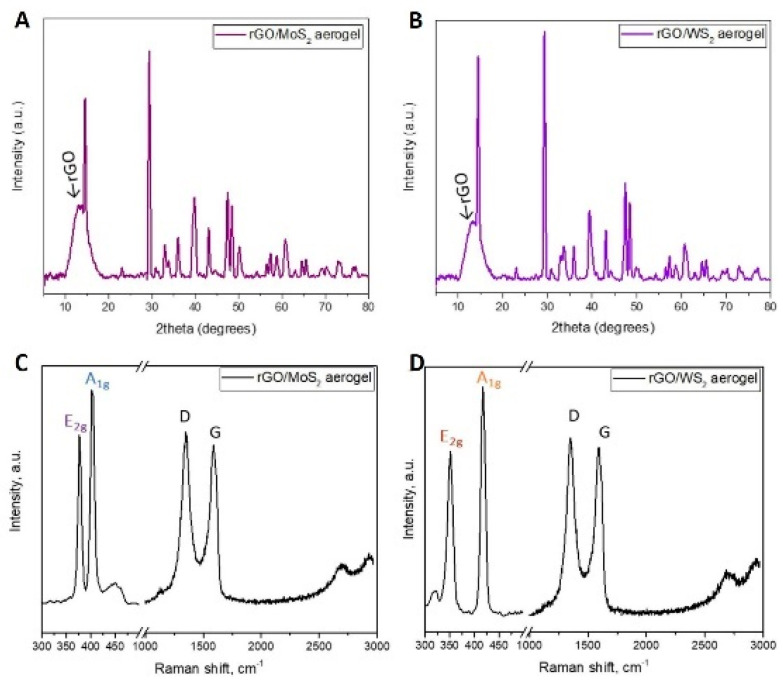
XRD and Raman spectra for the rGO/MoS_2_ *50/50* (**A**,**C**) and rGO/WS_2_ *50/50* (**B**,**D**) aerogels. The characteristic peaks of the 2D materials are highlighted in the Raman spectra.

**Figure 5 materials-17-03174-f005:**
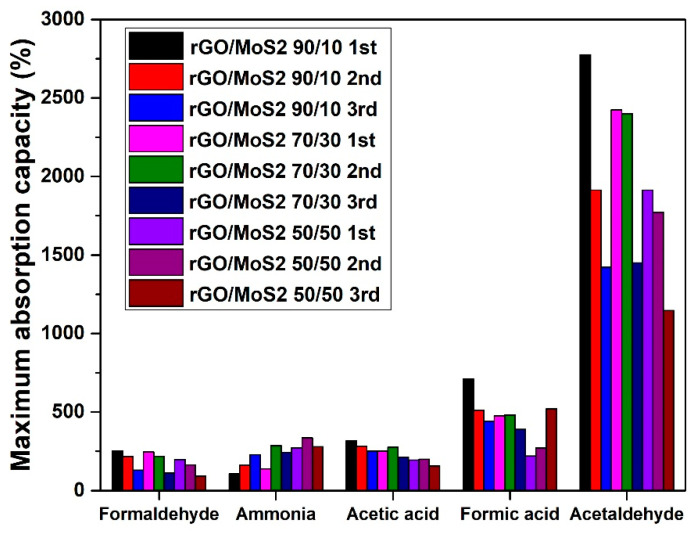
Maximum absorption capacities of rGO/MoS_2_ aerogels for three absorption-desorption cycles with VOCs.

**Figure 6 materials-17-03174-f006:**
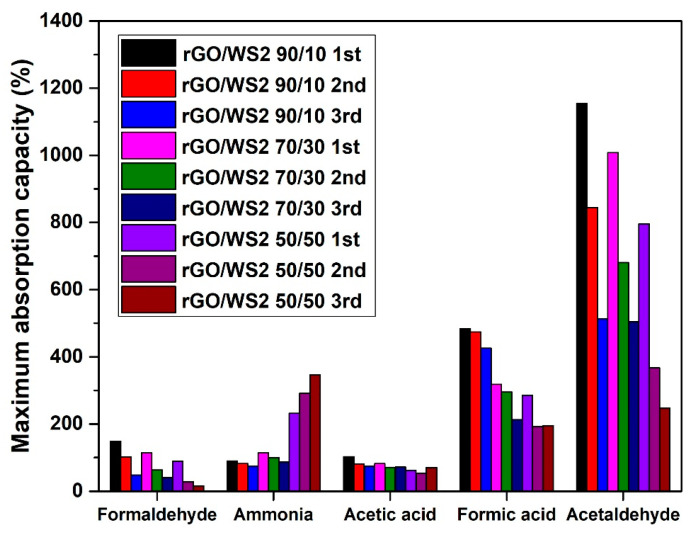
Maximum absorption capacities of rGO/WS_2_ aerogels for three absorption–desorption cycles with VOCs.

**Figure 7 materials-17-03174-f007:**
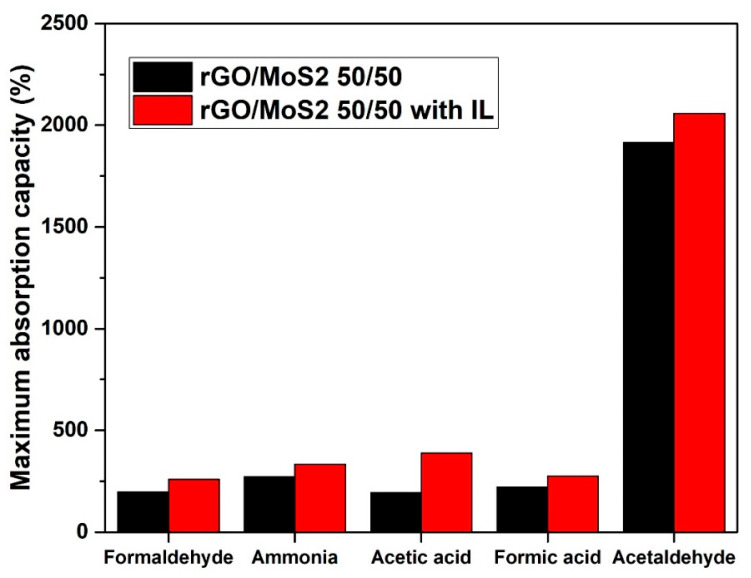
Effect of the IL on the maximum absorption capacities of rGO/MoS_2_ *50/50* aerogels.

**Figure 8 materials-17-03174-f008:**
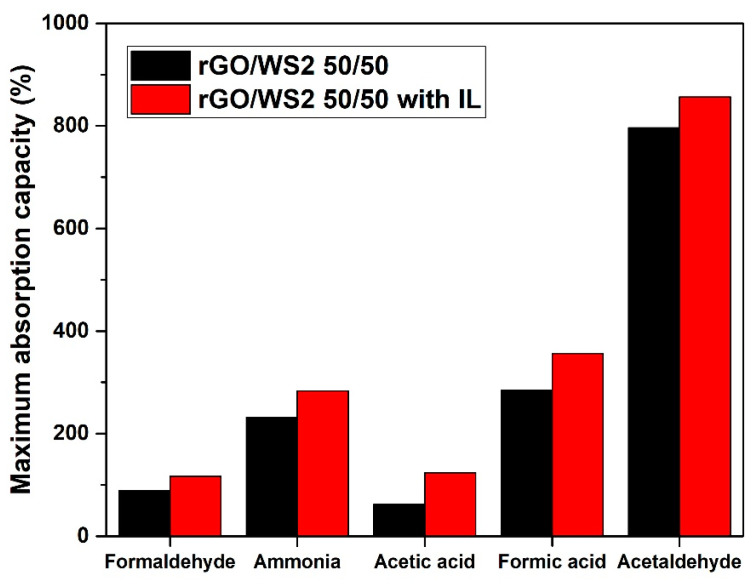
Effect of the IL on the maximum absorption capacities of rGO/WS_2_ *50/50* aerogels.

**Figure 9 materials-17-03174-f009:**
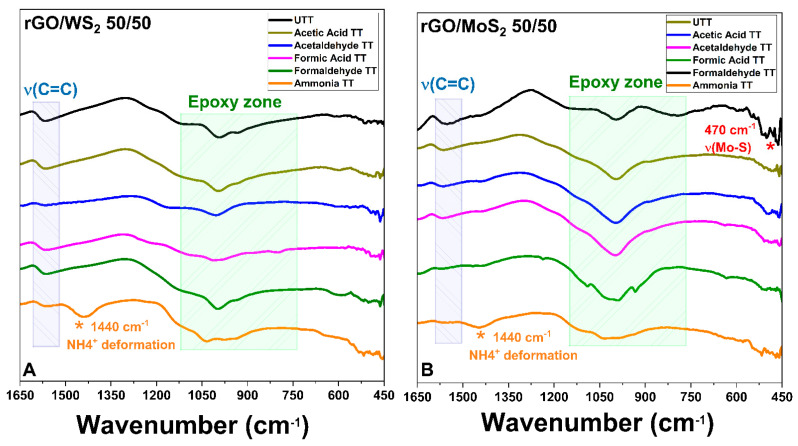
FTIR spectra of rGO/TMD ((**A**) for rGO/WS_2_ *50/50* and (**B**) for rGO/MoS_2_ *50/50*) after absorption tests with acetic acid (gold line), acetaldehyde (blue line), formic acid (pink line), formaldehyde (olive line), and ammonia (orange line). The spectra of aerogels as prepared (UTT), after exposure to the pollutant for 168 h (TT), and after the desorption test are compared.

**Figure 10 materials-17-03174-f010:**
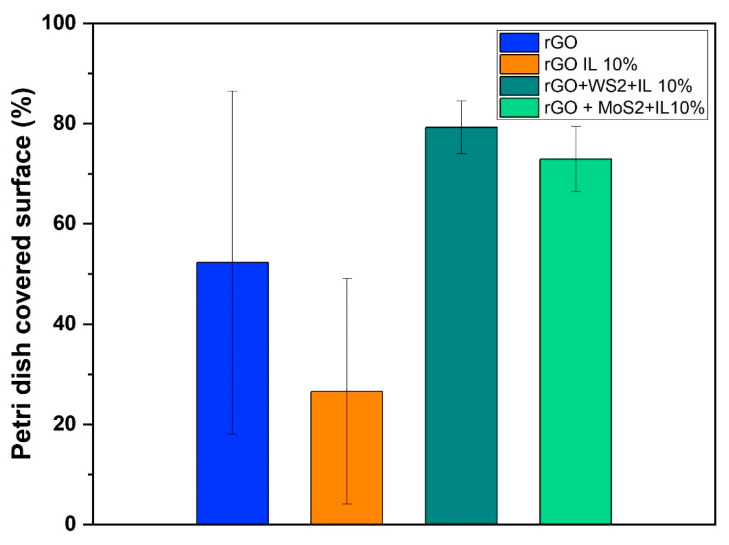
Graph of the percentage of the Petri dishes colonized by the fungus.

**Figure 11 materials-17-03174-f011:**
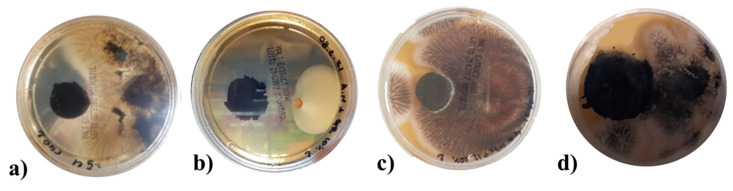
Representative pictures of the Petri dishes colonized by the fungus *Aspergillus niger* in the presence of (**a**) rGO, (**b**) rGO + IL 10%, (**c**) rGO/WS_2_ + IL 10%, and (**d**) rGO/MoS_2_ + IL 10% aerogels.

**Table 1 materials-17-03174-t001:** Percentage of Petri dish surface covered by the fungus.

	Average %	St.D.
rGO	52.27	34.20
rGO + IL 10%	26.56	22.48
rGO/WS_2_ + IL 10%	79.28	5.29
rGO/MoS_2_ + IL 10%	72.93	6.50

## Data Availability

The original contributions presented in the study are included in the article/Appendix A, further inquiries can be directed to the corresponding authors.

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
