# Peer review of "Antifungal Hybrid Graphene–Transition-Metal Dichalcogenides Aerogels with an Ionic Liquid Additive as Innovative Absorbers for Preventive Conservation of Cultural Heritage"

_materials, 2024, doi:10.3390/ma17133174_

Round 1
Reviewer 1 Report
Comments and Suggestions for Authors
The research presented in the manuscript "Antifungal Hybrid Graphene/Transition-Metal Dichalcogenides Aerogels with an Ionic Liquid Additive as Innovative Absorbers for Preventive Conservation of Cultural Heritage" aims to solve the problem of removing harmful substances that may lead to degradation of works of art from the air. The proposed solution for this problem relies on the preparation of three-dimensional aerogels from graphene and transition-metal dichalcogenides that could absorb certain harmful chemicals like volatile organic compounds (VOC) and ammonia. The results showed that the proposed material outperforms currently used absorbers for ammonia, formic acid, acetic acid formaldehyde, and acetaldehyde. In my opinion, this study is interesting and provides some useful results for the material design community, and could be accepted after addressing the following points:
1. Authors stated that "the incorporation of TMDs in rGO aerogels favored the selective absorption of ammonia" and "the most efficient ammonia absorbers were the hybrid aerogels with the highest amount of MoS2 and WS2". Could authors offer possible reasons for this result?
2. Authors also claimed that "Owing to the existence of a lone pair electrons, ammonia behaves as a charge donor to provide electrons to get physiosorbed on TMDs. Due to the strong electronegativity of the elemental sulfur layer of TMDs layers,bNH3 molecules are easily absorbed to the edge sites of the few-layered and bulk TMDs and transfer electrons to the TMDs layers to form NH4+. Thus, NH4+ ions adhere to the edges of two adjacent elemental sulfur layers and expand the interlayer space.". Is there any experimental evidence for that? If there are, these claims should be clearly put into the context.
3. Although the authors performed their research in the context of the Preventive Conservation of Cultural Heritage, I believe that these results are of much general importance, including for the chemical industry and research laboratories, and that should be stated in the text.
Reviewer 2 Report
Comments and Suggestions for Authors
lines 74-88 - add which fabrics are most affected by microorganisms and which other microorganisms colonize; state how mold can be prevented
lines 122-136 - list the pros and cons for the processes
lines 157, 159, 222 and 396 - "w/w" should be written in italics
lines 173-177 - it is necessary to give the range of wave numbers within which the measurement was made
line 190 - "λi and Q" should be written in italics
line 215 - "petri dishes" should be written with capital letter
lines 224, 289, 297, 305, 346, 348, 350, 357 - The references are not cited appropriate.
Figure 2 and Figure 3- should be enlarged and sharpened
Figure 6 - set the y-axis range to 1400
Figure 7 - set the y-axis range to 2500
Figure 9 - the left image is blurry, it should be sharpened
Figure 10 - this graph is arranged differently from the others, so put the legend inside the chart area and add a border for the chart drawing area
References - The references are not cited appropriate neither in the text nor in the list. Please check and correct.
Comments on the Quality of English Language
This work from Gorgolis et al. entitled Antifungal Hybrid Graphene/Transition-Metal Dichalcogenides Aerogels with an Ionic Liquid Additive as Innovative Absorbers for Preventive Conservation of Cultural Heritage is original work and it is appropriatefor publication after minor revision.
Reviewer 3 Report
Comments and Suggestions for Authors
In this manuscript, graphene and transition-metal dichalcogenides were combined to construct 3D aerogels for adsorbing/absorbing some harmful VOC substances in art micro-environments. Also, the addition of ionic liquid was investigated in terms of its effect on adsorption/absorption process. In general, the work is interesting both in material design and in oriented application. The manuscript is acceptable even at its current status, hopefully after minor revisions of some details as follows:
1. The reviewer have noticed the authors use both absorb/adsorb or absorption/adsorption for the VOC capturing process. For some discussions related to active sites, adsorb/adsorption should be used, though for some other discussions, absorb/absorption may be appropriate. The authors are suggested to recheck or reaffirm the use of these two different concepts;
2. In Line 208, the regeneration seems to be drying the adsorbent and removing the adsorbate. Please check the expression;
3. For comparison, ‘compared with’ seems more appropriate than ‘compared to’, as the authors seem to highlight the difference rather than the similarity;
4. The abbreviations TMD and TDM should be confirmed, as ‘TDM’ occurs several times, especially in the discussion of antifungal properties.
Comments on the Quality of English Language
Excellent English expression!
Reviewer 4 Report
Comments and Suggestions for Authors
In this manuscript, the authors reported the addition of TMD platelets into rGO aerogel for absorbing VOCs. In addition, the addition of IL decreases the growth of fungus. Although the idea is interesting, the conclusion is not well supported by the experimental results. Moreover, the investigation is not well designed. Therefore, I cannot recommend acceptance in its current form. I also have the following comments.
1. The MoS2 and WS2 bulk platelets were only sonicated for 10 min. In this case, they are still thick flakes or platelets. Why the authors added these thick platelets for preparing hybrid aerogels? Does the thickness or size influence the performance of hybrid aerogel?
2. The authors should provide the data of pure rGO aerogel to compare the effect of TMD adding into the aerogel on the VOCs absorption.
3. With increased amount of TMD, the absorption decreases. In this case, why the authors added the TMD?
4. As shown in Figure 5 and 6, the highest absorption occurred for CH3CHO. However, the authors discussed the absorption of ammonia with quite long sentences. It is strange to understand the point of this work.
5. As shown in Figure 7, the absorption decreases as the IL was added into the hybrid aerogels. Why they discuss this investigation?
6. The different absorption between rGO/MoS2 and rGO/WS2 aerogels is not well explained.
7. The resolution of Figure S1 and S2 is too low to observe them clearly.
8. They treated the samples by hairdryer for 24 h. This operation is not professional and serious. Put the sample into oven with vacuum should be better.
9. With the increased addition of TMD, what is the change of pore size and density of aerogel?
Comments on the Quality of English Language
Minor editing of English language required.
Round 2
Reviewer 4 Report
Comments and Suggestions for Authors
I have read the response from the authors. However, I don't think they treat my comments seriously. They haven't provided any additional experimental results to reply my comments. The conclusion is not well supported by the data shown in current version. Therefore, I cannot recommend publication in Materials. I also don't oppose acceptance if the editor considers to do it. BTW, I won't want to review it again if they don't reply the comments seriously.
Comments on the Quality of English Language
Minor editing of English language required.
Round 3
Reviewer 4 Report
Comments and Suggestions for Authors
Although I still not satisfied with the response from authors, I won't oppose acceptance now.